# Capillary pericytes mediate coronary no-reflow after myocardial ischaemia

**Fergus M O'Farrell[†], Svetlana Mastitskaya[†], Matthew Hammond-Haley[†], Felipe Freitas, Wen Rui Wah, David Attwell\***

Department of Neuroscience, Physiology and Pharmacology, University College London, London, United Kingdom

**Abstract** After cardiac ischaemia, a prolonged decrease of coronary microvascular perfusion often occurs even after flow is restored in an upstream artery. This 'no-reflow' phenomenon worsens patient prognosis. In the brain, after stroke, a similar post-ischaemic 'no-reflow' has been attributed to capillary constriction by contractile pericytes. We now show that occlusion of a rat coronary artery, followed by reperfusion, blocks 40% of cardiac capillaries and halves perfused blood volume within the affected region. Capillary blockages colocalised strongly with pericytes, where capillary diameter was reduced by 37%. The pericyte relaxant adenosine increased capillary diameter by 21% at pericyte somata, decreased capillary block by 25% and increased perfusion volume by 57%. Thus, cardiac pericytes constrict coronary capillaries and reduce microvascular blood flow after ischaemia, despite re-opening of the culprit artery. Cardiac pericytes are therefore a novel therapeutic target in ischaemic heart disease.

DOI: https://doi.org/10.7554/eLife.29280.001

## Introduction

Coronary heart disease is the leading cause of mortality worldwide, causing 7.4 million deaths in 2015. To treat cardiac ischaemia, primary percutaneous coronary intervention is used to re-open the culprit coronary artery, but this does not guarantee reperfusion of the downstream capillaries supplying the myocardium (*Krug et al., 1966*; *Kloner et al., 1974*). A lack of capillary reperfusion - 'no-reflow' - affects up to 50% of patients (*Niccoli et al., 2009*), and predicts a raised prevalence of deleterious complications including congestive heart failure, malignant arrhythmias and cardiac death (*Niccoli et al., 2009*; *Wu et al., 1998*; *Morishima et al., 2000*; *Ørn et al., 2009*; *Kloner, 2011*). No-reflow has been attributed to swollen endothelial cells and/or cardiomyocytes compressing the lumen of capillaries (*Kloner, 2011*), or to leukocytes adhering to and plugging capillaries (*Engler et al., 1983*). However, although depleting blood of granulocytes reduces no-reflow (*Engler et al., 1986*), leukocyte adhesion to capillaries in reperfused hearts is modest compared to the adhesion seen in post-capillary venules (*Habazettl et al., 1999*). No treatment exists for coronary no-reflow after decades of investigation, even though it predicts a worse outcome after myocardial ischaemia (*Niccoli et al., 2009*; *Wu et al., 1998*; *Morishima et al., 2000*; *Ørn et al., 2009*; *Kloner, 2011*). We therefore sought an alternative explanation for coronary capillary narrowing, which might trap leukocytes or red blood cells physically, based on recent advances in our understanding of the similar no-reflow phenomenon which occurs after brain ischaemia (*O'Farrell and Attwell, 2014*).

In the brain and retina, blood flow is partly regulated by contractile pericytes located on capillaries (*Hirschi and D'Amore, 1996*), which respond to vasoactive messengers released from active neurons and astrocytes by altering their tone, thus altering capillary diameter and blood flow (*Peppiatt et al., 2006*; *Puro, 2007*; *Hall et al., 2014*; *Biesecker et al., 2016*; *Duan et al., 2016*; *Kisler et al., 2017*). Furthermore, ischaemia, spinal cord injury and epilepsy lead to a contraction of

**\*For correspondence:**
d.attwell@ucl.ac.uk

[†]These authors contributed equally to this work

**Competing interests:** The authors declare that no competing interests exist.

**eLife digest** Heart attacks occur when one of the arteries supplying blood to the heart muscle becomes blocked, usually by a blood clot. Doctors unblock the artery and insert an expanding metal cage called a stent to keep it unblocked. This restores blood flow through the artery. Unfortunately, blood flow often does not return to smaller downstream blood vessels called capillaries. This can lead to further damage to the heart.

Scientists have not been able to find a way to reliably open up those capillaries after a heart attack because it is not clear exactly what is keeping them closed. Muscle-like cells called pericytes, which wrap around the capillaries, are one possible culprit for the blockages. Pericytes narrow capillaries in the brain after stroke in animal experiments. These cells are also present on heart capillaries, but scientists do not know much about them.

Now, O'Farrell, Mastitskaya, Hammond-Haley et al. show that pericytes are partly responsible for limiting blood flow in capillaries after a heart attack in rats. In the experiments, blood flow through an artery feeding the hearts of anaesthetized rats was restricted, simulating a heart attack. After the blood flow was later restored, 40% of the animal's capillaries remained blocked. Many blockages occurred near pericytes that had narrowed the capillary preventing blood flow. Treating the rats with a drug called adenosine, which relaxes the pericytes, reduced capillary blockages and increased blood flow in the heart.

Although adenosine could help to restore blood flow in the capillaries after a heart attack, it may also relax muscles around arteries and lower blood pressure, and so it may not be an ideal treatment. More studies are needed to determine whether drugs that target only the pericytes could complement existing heart attack treatments that unblock the arteries. If these studies are successful, pericyte-targeting drugs might prevent serious complications after a heart attack, including heart failure, heart rhythm abnormalities and future heart attacks.

DOI: https://doi.org/10.7554/eLife.29280.002

pericytes, causing constriction of capillaries (*Hauck et al., 2004*; *Peppiatt et al., 2006*; *Yemisci et al., 2009*; *Hall et al., 2014*; *Li et al., 2017*; *Leal-Camanario et al., 2017*), and for ischaemia this is followed by the pericytes dying in rigor (*Hall et al., 2014*) which hinders subsequent capillary dilation. Pericytes also exist in the heart (*Tilton et al., 1979*), where they are the second most numerous cell type after endothelial cells (*Nees et al., 2012*). They are poorly understood (only 1 in 5000 cardiac papers in PubMed mentions pericytes): it is debated whether they are contractile (*Tilton et al., 1979*; *Joyce et al., 1985*) and attention has focused on other possible functions for them including angiogenesis, immune defence, haemostasis and cardiac regeneration (*Nees et al., 2013*; *Chen et al., 2015*). Here we show that ischaemia-induced constriction of coronary capillaries by pericytes contributes to no-reflow after cardiac ischaemia.

## Results

### Pericytes associate with capillaries and sympathetic axons, and express actin

The left ventricular coronary capillary bed comprises an array of parallel capillaries linked by occasional connector capillaries running roughly orthogonal to the main direction of flow (*Figure 1A*). Labelling capillaries with FITC- or Alexa647-isolectin B$_4$, and pericytes in rat with antibody to NG2 (*Figure 1A*) or PDGFRβ (*Figure 1C*), or in mouse with transgenic expression of DsRed under the NG2 promoter (*Figure 1C*), showed that the great majority of coronary capillaries were contacted by pericytes. Pericytes differ from vascular smooth muscle cells in that their somata are spatially separated along the capillary, located either with a bump-on-a-log appearance on the straight parts of capillaries, or at the branch points of capillaries (*Attwell et al., 2016*). More of the parallel capillaries than of the connector capillaries linking adjacent parallel capillaries received pericyte contacts (92% vs 66%, p=2.2×10$^{-5}$, *Figure 1B*).The great majority of pericytes labelled for both NG2 and PRGFRβ (*Figure 1D*), although a small fraction (6.4 ± 0.2%) of PDGFRβ-expressing pericytes did not express NG2 (p=2.3×10$^{-7}$). Overall there is one pericyte every ~60 μm along rat coronary capillaries,

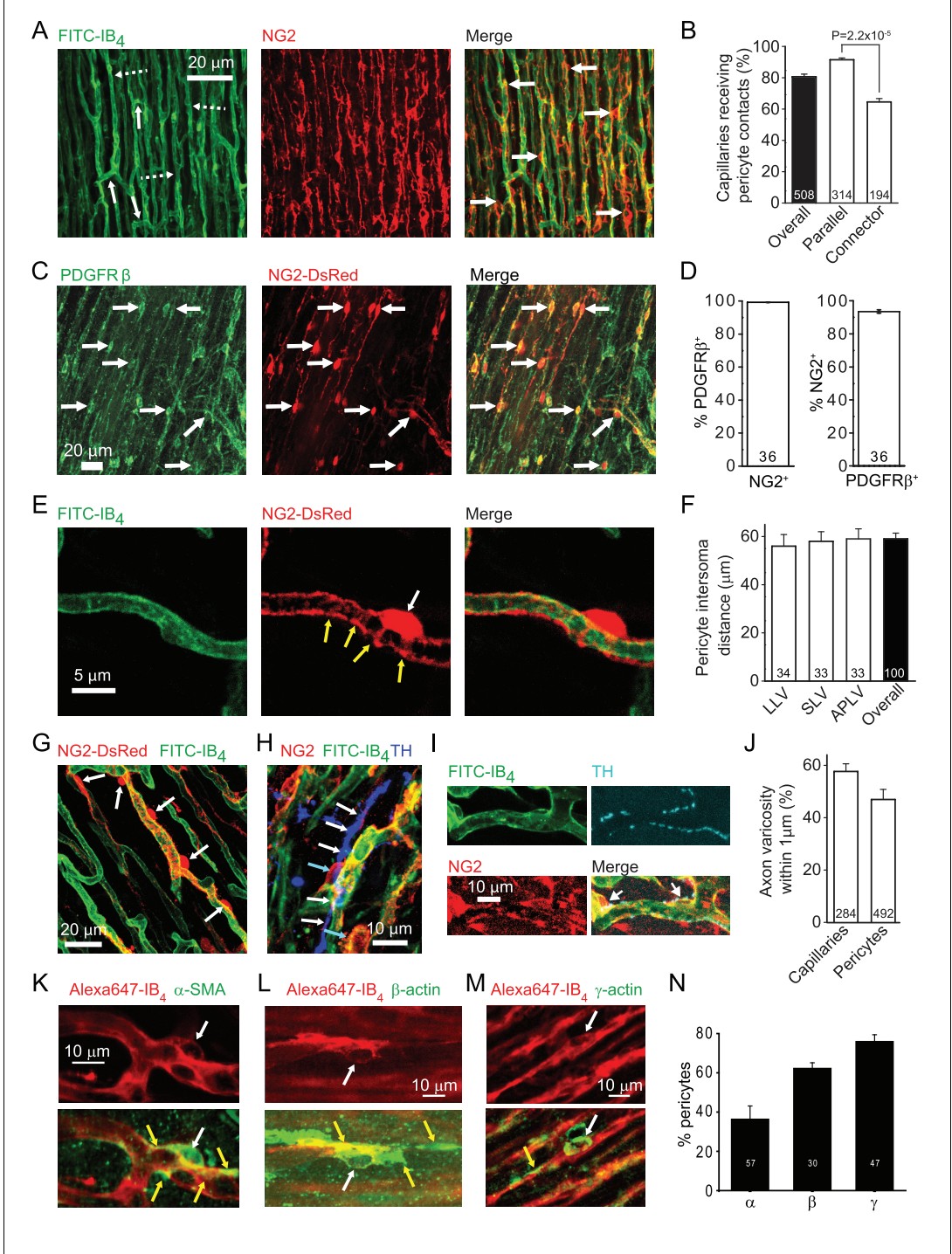

**Figure 1.** Cardiac pericyte morphology is appropriate for regulating capillary diameter. (**A**) Coronary capillaries in rat left ventricle (dashed arrows in left panel show longitudinal capillaries, solid arrows show connector capillaries) labelled for basement membrane (FITC-isolectin B$_4$) and for pericytes (example somata labelled with arrows in right panel) with antibody to the proteoglycan NG2. (**B**) A larger percentage of parallel capillaries (in rat,) receive pericyte contacts (<1 μm away) than do connector capillaries running between the parallel capillaries. (**C**) Mouse pericytes labelled with antibody to PDGFRβ (example somata labelled with arrows) and with NG2-DsRed. (**D**) Percentage of 2874 NG2-expressing pericytes (left) that also express PDGFRβ and of 3047 PDGFRβ—expressing pericytes (right) that also expressed NG2 (averaged over 36 confocal stacks each, from 4 mice, 2–9 months old) in the left ventricle of NG2-dsRed mice. (**E**) Pericyte in NG2-DsRed mouse showing soma (white arrow) and circumferential and longitudinal processes (yellow arrows). (**F**) The mean rat pericyte inter-soma distance is similar in the lateral wall of the left ventricle (LLV), the septal wall of the left ventricle (SLV), and anterior and posterior walls of the left ventricle (APLV). (**G**) Mouse pericyte circumferential processes can extend over much of the

*Figure 1 continued on next page*

Figure 1 continued

capillary surface between pericyte somata. (H) Labelling of tyrosine hydroxylase (blue) shows a close association of sympathetic axons (white arrows) with rat pericytes (cyan arrows). (I) At higher magnification, tyrosine hydroxylase labelled axon varicosities (putative transmitter release sites, cyan) can be seen apposed to pericyte somata and processes. (J) Both capillaries and pericytes (including soma and processes) frequently have sympathetic axon varicosities within 1 μm (in rat). (K–M) Examples of pericytes labelled with Alexa647-isolectin B$_4$ (red) that also labelled (green) for α-SMA (K), β-actin (L) or γ1-actin (M). White arrows indicate somata; yellow arrows indicate actin-labelled processes. (N) Percentage of pericytes expressing the 3 actin isoforms in three hearts. Data are mean ± s.e.m. Numbers on bars are of capillaries (panels B, J), images (D), intersoma distances (F) or pericytes (J, N).

DOI: https://doi.org/10.7554/eLife.29280.003

independent of location in the left ventricle (*Figure 1F*) although there is variability around this mean distance (*Figure 1A,G*).

Pericytes extend circumferential processes (*Figure 1E,G*), the contraction of which would directly alter capillary diameter. Contraction of their longitudinal processes could alter the stiffness of the capillary wall and regulate its deformability by passing blood cells. Either of these mechanisms could regulate capillary blood flow. Indeed, varying degrees of contraction of these pericytes is a plausible explanation for the large variance of capillary blood transit time in the coronary circulation (*Rose and Goresky, 1976*), which has a significant effect on oxygen extraction by the myocardium (*Ostergaard et al., 2014*). We found that approximately 50% of pericytes are located close to varicosities of tyrosine hydroxylase expressing sympathetic axons (*Figure 1H–J*), suggesting the possibility of noradrenergic regulation of pericyte tone.

Pericytes are conventionally assumed to constrict capillaries using α-smooth muscle actin (α-SMA: *Joyce et al., 1985*; *Skalli et al., 1989*), but variability in the labelling observed for α-SMA and data showing expression of other actin isoforms in pericytes has led to a suggestion, for CNS pericytes, that γ-actin might instead be the relevant actin isoform (*DeNofrio et al., 1989*; *Grant et al., 2017*). We therefore examined antibody labelling for α-SMA, β-actin and γ-actin (*Figure 1K–M*). α-SMA labelling occurred in 36.4 ± 6.7% of 57 pericytes (*Figure 1N*), and was rarer than labelling for β-actin (62.3 ± 2.8% of 30 pericytes) or γ-actin (76.0 ± 3.4% of 47 pericytes). α-SMA-expressing pericytes were most commonly observed in capillaries closer to the arteriole end of the capillary bed, while β- and γ-actin were seen in pericytes across the capillary bed.

## No-reflow occurs after coronary ischaemia

To examine the possible role of contraction of pericytes in ischaemic pathology, we occluded the left anterior descending (LAD) coronary artery for 45 mins, depriving the anterior wall of the left ventricle and part of the right ventricle of blood (see Materials and methods). We then removed the occlusion to allow reperfusion for 15 mins, so that any rapidly reversible obstruction of blood vessels produced by ischaemia would be removed, leaving only long-lasting vessel obstruction contributing to the no-reflow phenomenon. At the end of this period FITC-albumin in gelatin was perfused to visualise vessels where flow was present, and the tissue was sectioned and labelled with isolectin B$_4$ conjugated to Alexa Fluor 647 to visualise non-perfused vessels. In control (sham LAD artery occlusion) hearts, perfusion was visible throughout the whole cross section of the left and right ventricles (*Figure 2A*). The perfused blood volume per unit area (assessed from the mean FITC-albumin intensity in regions of interest) was fairly uniform around the left ventricle (*Figure 2C*), although somewhat higher in the posterior wall (regions of interest (ROIs) 4–6 of *Figure 2C*). In contrast, occlusion and reperfusion of the LAD artery resulted in microvascular perfusion being greatly reduced after the period of ischaemia in half of the left and right ventricles (*Figure 2B*), as quantified for the left ventricle in *Figure 2C* (reduced by 49% compared to sham-operated animals in ROIs 7–10: p=0.0004).

## Pericytes constrict capillaries after ischaemia

Higher magnification images revealed that in control hearts only 3% of capillaries were blocked (i.e. not perfused by FITC-albumin) in the left ventricle. In contrast, LAD artery occlusion and reperfusion increased this to 40% in the affected area (*Figure 2D*). Some capillaries were completely perfused and some were completely unperfused throughout the imaged area, while some capillaries showed an abrupt cessation of perfusion (*Figure 3A–F*), with a profound decrease of FITC-albumin intensity that occurred over a few microns (*Figure 3G*). Since leukocytes are both larger and less deformable

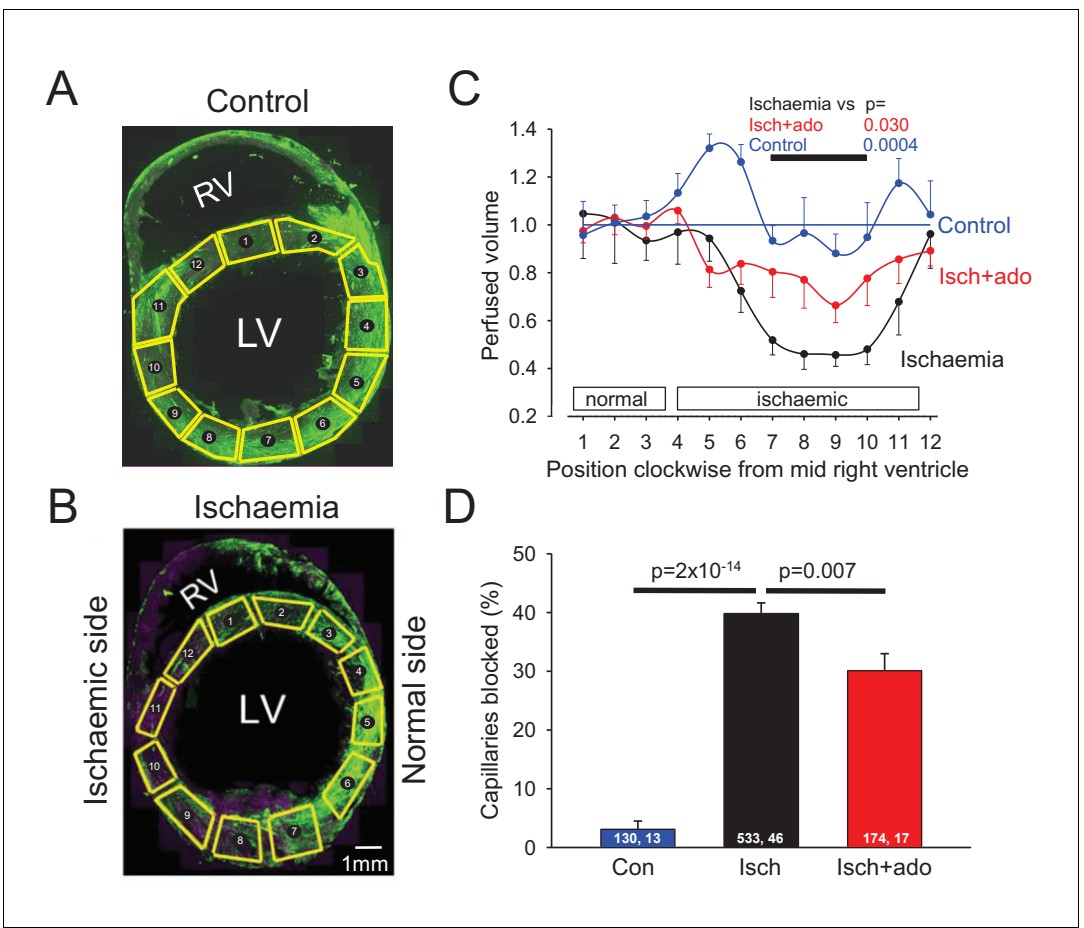

**Figure 2.** Ischaemia and reperfusion lead to no-reflow mediated by capillary block. (A, B) Low power view of sham-operated heart (A) and a heart after LAD coronary artery occlusion and reperfusion (B), with perfusion volume assessed as intensity of FITC-albumin (green). In (B) vessels are also labelled with isolectin $B_4$ - Alexa Fluor 647 (purple) to define location of unperfused tissue. Regions of interest (ROIs) for analysing the intensity of FITC-albumin fluorescence are shown in yellow. (C) Perfused volume (assessed from mean FITC-albumin intensity), in ROIs indexed with numbers starting at the interventricular septum and proceeding clockwise around the left ventricle (as seen from above), for five sham-operated hearts (control), six hearts made ischaemic and reperfused (ischaemia), and eight hearts made ischaemic and exposed to adenosine starting 5 min before reperfusion (isch + ado). (D) Percentage of capillaries blocked in the anterior wall of the left ventricle for the three experimental conditions (numbers on bars are of 'capillaries examined, image stacks examined'). Data are mean ± s.e.m. P values are corrected for multiple comparisons.

DOI: https://doi.org/10.7554/eLife.29280.004

than erythrocytes (*Schmid-Schönbein et al., 1981*; *Downey et al., 1990*; *Komatsu et al., 1990*; *Doerschuk et al., 1993*), it seemed more likely that leukocytes rather than red blood cells would get stuck at capillary regions of reduced diameter. Surprisingly, labelling with antibody to neutrophil elastase or ICAM-1 revealed no leukocytes lodged at 46 blockage sites examined (although, as a positive control, they were seen outside vessels, *Figure 3K*, usually post-capillary venules). Similarly, labelling for the erythrocyte protein glycophorin A revealed red blood cells (*Figure 3L*) associated with only a small percentage of blockage sites (18% of 44 blockages), and even where red blood cells were trapped at capillary constrictions it did not always lead to a block of blood flow (as shown by FITC-albumin passing the red blood cells). However, examining NG2 labelling at the sites of block revealed that many blockages occurred close to pericytes, in some cases with pericyte processes appearing to visibly constrict the capillary at the block location (*Figure 3A–C*).

To assess rigorously whether block occurred disproportionately close to pericytes, we measured the distance of 42 blockages to the nearest pericyte soma. The cumulative probability distribution of

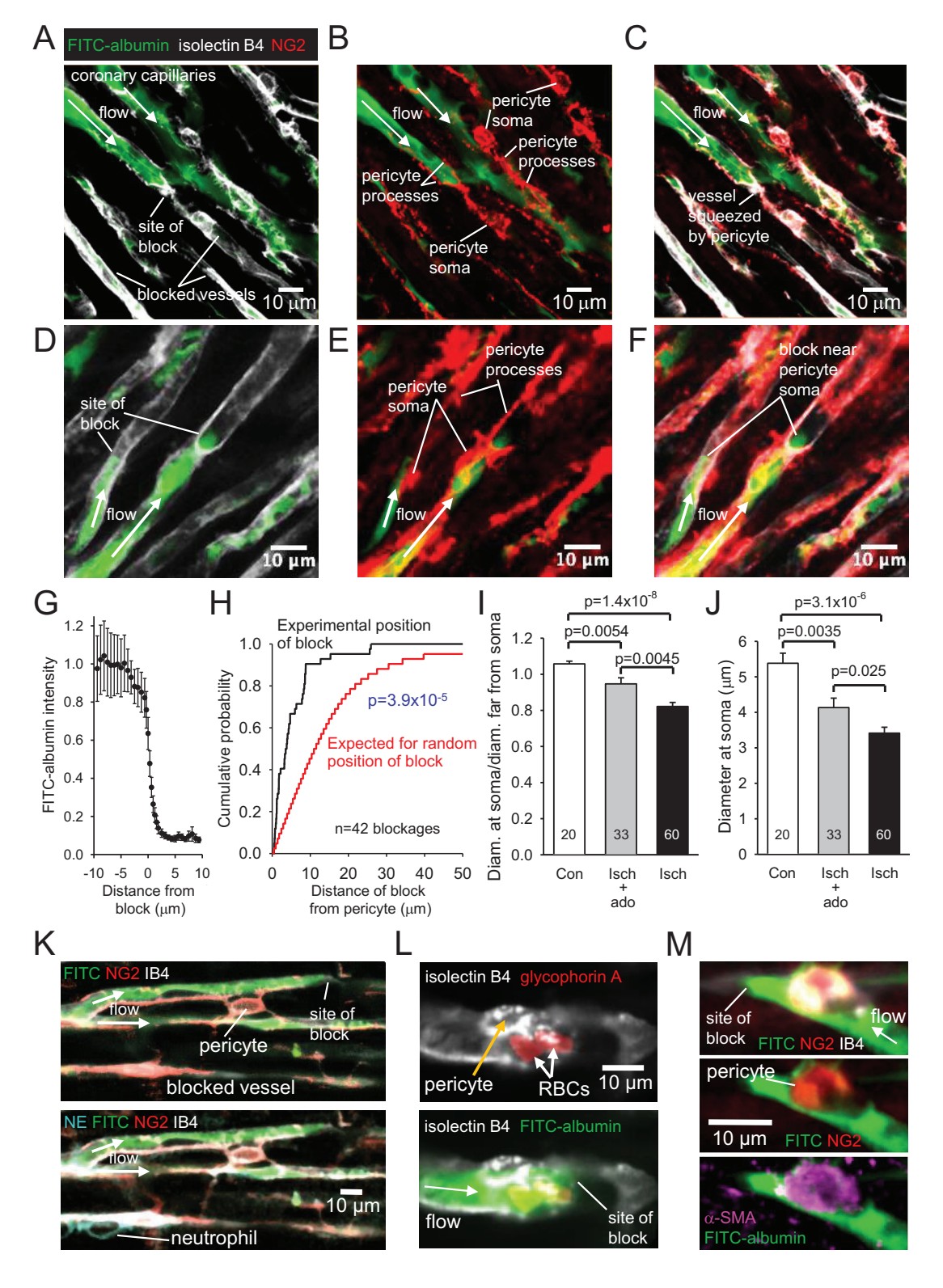

**Figure 3.** No-reflow reflects blockage by pericyte constriction. (A) Image of perfused and non-perfused capillaries in post-ischaemic left ventricle. Isolectin $B_4$ labelling (white) defines positions of all vessels, while FITC-albumin labelling (green) shows vessels that are perfused. Bottom left capillary is completely non-perfused; top green capillary is fully perfused; lower green capillary is blocked halfway across the image. (B, C): NG2-labelling of pericytes (B) and merge (C) of the images (A) and (B) show pericyte processes constricting vessel at block site. (D–F) Another example set of images as *Figure 3 continued on next page*

*Figure 3 continued*

in (**A–C**), showing two capillaries blocked near pericyte somata. (**G**) Normalised intensity of (background-subtracted) FITC-albumin (green) labelling along the centre of the capillary lumen across 20 block sites. (**H**) Cumulative probability distribution for the distance from capillary blockage sites to the nearest pericyte soma (black) and for the position expected (see *Figure 3—figure supplement 1*) if blocks occurred at positions independent of pericyte locations (significantly different, p=$3.9 \times 10^{-5}$). Comparing the experimental distribution with a theoretical distribution increasing linearly to one at a distance of 30 µm (see Image Analysis in Materials and methods) also showed a significant difference (p=$7.6 \times 10^{-8}$). (**I**) Ratio of capillary diameter at pericyte somata to the diameter at positions ~ 10 µm upstream after ischaemia, after ischaemia with adenosine (ado), and for sham-operated hearts (Con). (**J**) Diameter at pericyte somata after ischaemia, after ischaemia with adenosine, and for sham-operated hearts (Con) (all pericyte locations were measured, not just those associated with capillary blockages, for which the mean diameter after ischaemia was smaller: 3.19 ± 0.24 µm, n = 30). (**K**) Capillary blockage in an area of the heart with a neutrophil (labelled for neutrophil elastase, NE, bottom left) present outside the capillaries. (**L**) Pericyte (labelled with isolectin B$_4$) near a blockage constricting a vessel with two red blood cells (RBCs, labelled for glycophorin A) trapped in the constriction. (**M**) Blockage-associated pericyte (labelled for NG2) that also labels for α-SMA. Numbers on bars are of pericytes. Data are mean ± s.e.m. P values in I-J are from Mann-Whitney tests and are corrected for multiple comparisons.

DOI: https://doi.org/10.7554/eLife.29280.005

The following figure supplement is available for figure 3:

**Figure supplement 1.** Calculation of probability distribution for distance from a randomly placed block to a chosen pericyte soma.

DOI: https://doi.org/10.7554/eLife.29280.006

this distance showed the median distance to be 3.6 µm (*Figure 3H*). In contrast, when we used the same 42 images to calculate what the average distance of a blockage to the nearest pericyte soma would be if blockages were randomly placed on the blocked capillaries in the images (see *Figure 3—figure supplement 1*), the cumulative probability distribution for that distance predicted a median distance of 11.6 µm (*Figure 3H*; significantly different, p=$3.9 \times 10^{-5}$). Thus, ischaemia-evoked capillary block occurs disproportionately close to pericytes.

To verify that pericytes constrict coronary capillaries in ischaemia, we measured the diameter of the FITC-albumin labelled lumen at pericyte somata, and at a distance 10 µm upstream of the soma (since most circumferential processes of pericytes are located less than 10 µm from the soma). In sham-operated animals, the ratio of the diameter at the soma to that upstream was 1.058 ± 0.015 (n = 20, significantly greater than one, p=0.001, *Figure 3I*), consistent with the presence of the pericyte soma inducing growth of the capillary lumen, as found previously for brain pericytes (*Hall et al., 2014*). In contrast, for capillaries after ischaemia, this ratio was 0.822 ± 0.022 (n = 60, significantly less, p=$1.4 \times 10^{-8}$, *Figure 3I*), implying that constriction occurs near the pericyte soma. The absolute diameter measured at pericyte somata was reduced by 37% after ischaemia and reperfusion (compared to sham-operated hearts, p=$3.1 \times 10^{-6}$, *Figure 3J*).

Pericytes near blockage sites were tested for labelling of the different actin isoforms mentioned above. All 4 such pericytes tested for α-SMA labelling exhibited labelling (*Figure 3M*), as did 5 out of 6 tested for β-actin, and all 6 tested for γ-actin. Since only a small fraction of pericytes may need to constrict a capillary to abolish its blood flow, further work is needed to determine which is the main actin isoform responsible for the ischaemia-evoked contraction of pericyte processes and reduction of blood flow.

Since cardiac catecholamine transporters reverse and release noradrenaline and adrenaline in ischaemia (*Lameris et al., 2000*), and noradrenaline acts on pericytes in the brain to constrict capillaries (*Peppiatt et al., 2006*), we assessed whether blocking adrenergic α$_1$ receptors reduced the percentage of capillaries blocked after ischaemia. We found no effect of injecting terazosin (0.5 mg/kg i.v., 5 mins before occluding the LAD artery, which produced a 20–33 mm Hg decrease of blood pressure), with 48.8 ± 3.7% of capillaries blocked in the presence of terazosin (in 13 images covering 383 capillaries in 2 hearts; not significantly different from the 44.7 ± 4.9% seen for 14 images covering 366 capillaries in 2 hearts in interleaved experiments without terazosin, p=0.52). We therefore sought an alternative pharmacological approach to reducing no-reflow after ischaemia.

## Adenosine reduces pericyte constriction and no-reflow

Adenosine has been suggested to reduce no-reflow after cardiac ischaemia, although its clinical utility remains uncertain (*Berg and Buhari, 2012*; *Su et al., 2015*). Adenosine is also thought to relax pericytes (*Matsugi et al., 1997*; *Li and Puro, 2001*; *Gaudin et al., 2014*). We infused adenosine

intravenously (at 0.5 mg/kg/min, similar to doses used previously to treat no-reflow in humans and other animals: see Materials and methods), from 5 mins before the end of ischaemia until 10 mins of reperfusion had occurred (see Materials and methods), and examined its effect on capillary block. Adenosine significantly reduced (p=0.03) the decrease of coronary perfusion seen after ischaemia (*Figure 2C*, the mean value in ROIs 7–10 in *Figure 2C* was increased by 57% compared to ischaemia without adenosine, and was not significantly different from that in control conditions, p=0.77). This increase of flow could reflect adenosine acting both on arteriolar smooth muscle and on pericytes. However, adenosine also reduced by one quarter the percentage of capillaries that were blocked after reperfusion, from ~40% to ~30% (p=0.007, *Figure 2D*). An analysis like that in *Figure 3H* showed that the remaining sites of block were still significantly associated with pericyte locations (p=0.001). Since the capillary blockages induced by ischaemia in the absence of adenosine are disproportionately associated with pericytes, and since the adenosine was only applied around the period of reperfusion, these data suggest that, at least in part, adenosine reduces no-reflow by reversing the constriction of pericytes that ischaemia induces. To test this hypothesis, we compared the diameter of capillaries at pericyte somata with the diameter 10 µm upstream of the soma and found that adenosine significantly (p=0.0045) reduced the constriction evoked by ischaemia at pericyte somata (*Figure 3I*), implying a specific effect on pericytes (rather than a general capillary dilation produced by upstream arteriole dilation). The absolute capillary diameter at pericyte somata after ischaemia was increased by 21% using adenosine (p=0.025, *Figure 3J*). Thus, adenosine decreases no-reflow by relaxing pericytes.

## Discussion

Our data suggest a novel therapeutic target for reducing no-reflow after cardiac ischaemia: during ischaemia pericytes constrict and block coronary capillaries, probably because pericyte $[Ca^{2+}]_i$ rises when ion pumping stops in ischaemia. Three lines of evidence support this idea. First, sites of capillary blockage are disproportionately associated with pericyte locations (*Figure 3H*). Second, after ischaemia the diameter of capillaries is reduced specifically at pericyte somata (*Figure 3I,J*). Third, adenosine, which is thought to relax pericytes, increases the post-ischaemic diameter of capillaries at pericyte locations (*Figure 3I,J*) and reduces the percentage of capillaries that remain blocked when the upstream artery is reperfused after ischaemia (*Figure 2D*). Prevention or reduction of pericyte constriction should therefore reduce the flow impairment that causes significant functional impairment after removal of a thrombus in a coronary artery.

The most plausible explanation of our data is that, like pericytes in the CNS (*Peppiatt et al., 2006*; *Hall et al., 2014*), cardiac pericytes extend actomyosin-containing processes around coronary capillaries that, when their $[Ca^{2+}]_i$ rises and contraction is activated, reduce capillary diameter (see Supplementary Movie 1 of *Peppiatt et al., 2006*). Indeed, cardiac pericytes express both actin and myosin (*Joyce et al., 1985*; *Skalli et al., 1989*). It is likely that pericytes do not need to completely constrict a capillary in order to stop blood flow in that vessel, because a reduction in diameter (like the 37% in *Figure 3J*) may prevent the passage of leukocytes (*Engler et al., 1983*) or red blood cells (*Figure 3L*), causing a blockage that prevents the passage of both plasma and cells. The fact that we observed few cells in the capillary lumen at blockages may result from their displacement during the perfusion with FITC-albumin in gelatin, or may alternatively reflect capillary block occurring when constriction brings together the layers of glycocalyx (*Secomb et al., 1998*) on opposite sides of the capillary.

It is uncertain whether, over a long period, cardiac pericyte death in rigor occurs and contributes to generating a long lasting reduction of blood flow, as suggested for the CNS (*Hall et al., 2014*). Clearly not all pericytes die during the 45 min period of ischaemia used here (which in severe brain ischaemia kills ~50% of pericytes: *Hall et al., 2014*) because, if they did, then adenosine applied around the time of reperfusion would not be able to reduce the amount of capillary constriction and block occurring. However, adenosine only reduces the number of blocked capillaries by 25% (*Figure 3J*) and conceivably it would produce a greater restoration of flow if pericyte death were prevented.

It should be recognised that, as with all studies evoking ischaemia by transiently occluding a coronary artery in rodents, our work is only a partial simulation of human cardiac ischaemia. Typically, human myocardial infarct involves the slow build up of a plaque on an artery, which becomes

occluded by a thrombus after plaque rupture occurs. This is followed by mechanical removal of the clot (which may compound microvascular obstruction by embolising plaque and thrombus material), leaving an activated oedematous endothelium across which neutrophils invade the tissue (*Niccoli et al., 2009*; *2016*). Nevertheless, at short times after ischaemia, we expect the constriction that we observe of downstream capillary pericytes to occur also in human cardiac tissue, prolonging the period of ischaemia.

Previously, a combination of myocyte swelling, endothelial cell blebbing and leukocyte plugging have been proposed to block capillaries in ischaemia (*O'Farrell and Attwell, 2014*). Our demonstration that ischaemia selectively reduces capillary diameter near pericytes suggests that, at least at early times, the majority of capillary blockages in fact occur near pericytes that have constricted the capillaries. These data imply that pericytes are a novel drug target for treating the coronary no-reflow that often occurs after primary percutaneous coronary intervention. It would be interesting to test whether deletion of pericyte-capillary interactions mediated by the angiopoietin/TIE and PDGF/PDGFRβ pathways (*Ziegler et al., 2013*) affect the contractile properties of pericytes and their response to ischaemia. We used adenosine to relax pericytes (an effect verified from measurements of capillary diameter at pericyte locations in *Figure 3I–J*), but it may also act on arteriolar smooth muscle and lower blood pressure. To reduce no-reflow therapeutically by preventing pericyte constriction of coronary capillaries, it will be necessary to develop agents that act selectively on pericytes. Encouragingly, in the brain it has been shown that signalling pathways regulating pericyte constriction can differ from those regulating arteriole smooth muscle (*Mishra et al., 2016*). Our data also raise the possibility that other organs susceptible to ischaemic no-reflow pathology, such as the kidneys, muscle and skin, may also be damaged by a pericyte-mediated mechanism. Finally, given the proximity of cardiac pericytes to sympathetic terminals (*Figure 1H–J*), noradrenaline release may also regulate coronary blood flow at the capillary level in physiological conditions.

# Materials and methods

## Key resources table

| Reagent type (species) or resource | Designation | Source or reference | Identifiers | Additional information |
|---|---|---|---|---|
| *Rattus norvegicus* (Sprague Dawley, male) | Rat | UCL Biological Services | | |
| genetic reagent (*Mus musculus/spretus*, male and female) | NG2-DsRed mice | doi: 10.1242/dev.004895 | JAX 008241 | |
| antibody | anti-NG2 (rabbit polyclonal) | Merck Millipore | AB5320 | (1:200) |
| antibody | anti-NG2 (mouse monoclonal) | AbCam | ab50009 | (1:200) |
| antibody | anti-PDGF receptor beta (rabbit polyclonal) | Santa Cruz Biotechnology | sc-432 | (1:200) |
| antibody | anti-tyrosine hydroxylase (sheep polyclonal) | Merck Millipore | AB1542 | (1:500) |
| antibody | anti-alpha smooth muscle actin (rabbit polyclonal) | AbCam | ab5694 | (1:100) |
| antibody | anti-beta actin (mouse monoclonal) | Abbiotec | 251815 | (1:100) |
| antibody | anti-gamma actin (mouse monoclonal) | AbCam | ab123034 | (1:100) |
| antibody | anti-glycophorin A (mouse monoclonal) | AbCam | ab9520 | (1:2000) |
| antibody | anti-neutrophil elastase (goat monoclonal) | Santa Cruz | sc9521 | (1:50) |
| antibody | anti-ICAM1 (mouse monoclonal) | AbCam | Ab171123 | (1:100) |
| antibody | Alexa Fluor 405 goat anti-rabbit (polyclonal) | Life Technologies | A31556 | (1:500) |

*Continued on next page*

*Continued*

| Reagent type (species) or resource | Designation | Source or reference | Identifiers | Additional information |
|---|---|---|---|---|
| antibody | Alexa Fluor 555 donkey anti-rabbit (polyclonal) | Life Technologies | A31572 | (1:500) |
| antibody | Alexa Fluor 555 donkey anti-mouse (polyclonal) | Life Technologies | A31570 | (1:500) |
| chemical compound | isolectin $B_4$ - Alexa Fluor 647 | Molecular Probes | I32450 | (1:200) |
| chemical compound | isolectin $B_4$ - FITC | Sigma-Aldrich | L2895 | (1:200) |
| chemical compound | adenosine | Sigma-Aldrich | A9251 | |
| chemical compound | gelatin | Sigma-Aldrich | G2625 | 5% in PBS |
| chemical compound | FITC-albumin | Sigma-Aldrich | A9771 | 1:200 in 5% gelatin |
| chemical compound | DAPI | Molecular Probes | D1306 | |
| chemical compound | terazosin | Sigma | T4680 | |
| software | *Spike2* | www.ced.co.uk | | in vivo data acquisition |
| software | ImageJ | https://imagej.nih.gov/ij/ | | image analysis |
| software | OriginPro | www.originlab.com/Origin | | statistical analysis |

## Pericyte anatomy

Adult Sprague-Dawley rats or, to illustrate morphology, NG2-DsRed mice (*Zhu et al., 2008*) expressing the fluorescent protein DsRed in pericytes, of either sex, were sacrificed by UK government approved methods (anaesthetic overdose with 5% isoflurane in 100% $O_2$, followed by cervical dislocation). Hearts were harvested and immersion-fixed in ice cold 4% paraformaldehyde (PFA). Immunohistochemistry was performed on 150 µm thick transverse ventricular slices.

## Immunohistochemistry

Pericytes were labelled by expression of DsRed under control of the NG2 promoter (in mice), or with anti-NG2 (Merck Millipore AB5320 1:200 or AbCam ab50009 1:200) or anti-PDGFRβ (Santa Cruz Biotechnology sc-432 1:200) antibodies (in rats), and the capillary basement membrane was labelled with isolectin $B_4$ conjugated to Alexa Fluor 647 (Molecular Probes, I32450) or FITC (Sigma-Aldrich, UK, L2895) as described (*Mishra et al., 2014*). Pericyte association with sympathetic terminals was imaged using antibody to tyrosine hydroxylase (Merck Millipore AB1542 1:500). Z-stacks for cell counting were acquired using laser-scanning confocal microscopy (Zeiss LSM 700). Pericyte inter-soma distance was calculated between pairs of pericytes on capillaries within the same imaging plane. Antibodies to α-SMA, β-actin and $γ_1$ actin were AbCam ab5694 (1:100), Abbiotec 251815 (1:100) and AbCam ab123034 (1:100). Red blood cells were labelled with antibody to glycophorin A (AbCam ab9520, 1:2000). Neutrophils were labelled with antibodies to neutrophil elastase (Santa Cruz, sc9521 (M-18), 1:50) or ICAM1 (Abcam, Ab171123, 1:100).

## Animal preparation for ischaemia experiments

Adult male Sprague-Dawley rats (200–220 g) were anaesthetized with pentobarbital sodium (induction 60 mg/kg i.p.; maintenance 10–15 mg/kg/h i.v.). The right carotid artery and jugular vein were cannulated for measurement of arterial blood pressure and administration of anaesthetic and tested substances, respectively. Stable levels of blood pressure and heart rate were maintained, and adequate anaesthesia was monitored by the absence of a withdrawal response to a paw pinch. The trachea was cannulated, and the animal was mechanically ventilated with room air using a positive pressure ventilator (tidal volume of 1 ml/100 g of body weight, ventilator frequency ~60 strokes $min^{-1}$). $PO_2$, $PCO_2$ and pH of the arterial blood were measured regularly and, if required, ventilation was adjusted to maintain values within their physiological ranges. Arterial BP and a standard lead II ECG were recorded using a Power1401 interface and *Spike2* software (Cambridge Electronic Design). Body temperature was maintained at 37.0 ± 0.5°C with a servo-controlled heating pad.

## Myocardial ischaemia and reperfusion

The heart was exposed via a left thoracotomy and a 4–0 prolene suture was passed around the left anterior descending (LAD) coronary artery to induce a temporary occlusion as previously described (*Mastitskaya et al., 2012*; *Basalay et al., 2016*). The animals were subjected to 45 min of LAD artery ligation, followed by 15 min of reperfusion. Successful LAD occlusion was confirmed by paling of the myocardial tissue distal to the suture, elevation of the ST-segment in the ECG, and an immediate 15–30 mm Hg fall in the ABP. In rats LAD occlusion deprives of blood not only the anterior wall of the left ventricle but also part of the right ventricle (*Samsamshariat and Movahed, 2005*), as seen in *Figure 2*. Control (sham operated) animals underwent the same procedures, except that after the suture was passed around the LAD coronary artery it was not drawn tight to occlude the vessel. Ischaemia and sham animals were alternately interleaved.

In some experiments *i.v.* adenosine (Sigma-Aldrich, St. Louis, Missouri) was administered (with the aim of relaxing pericytes), starting 5 min before the end of ischaemia and continuing until 10 min of reperfusion (0.5 mg/kg/min in saline, 2 mg/ml, cumulative dose 7.5 mg/kg), which lowered the blood pressure reversibly by 10–15 mm Hg (10%). Control animals received vehicle (saline) infusion. For comparison, in cats 0.5 mg/kg/min is approximately the upper limit (*Portellos et al., 1995*) for restricting the adenosine-induced fall of blood pressure to less than 10%, in rats a cumulative dose of 3 mg/kg reduced infarct size by 31% (*Shafy et al., 2012*), and in humans infusing 0.07 mg/kg/min for 3 hr during reperfusion (a cumulative dose of 12.6 mg/kg) reduced infarct size by 57% in the AMISTAD-II trial (*Ross et al., 2005*), although the REFLO-STEMI trial found no beneficial effect of adenosine (*Nazir et al., 2016*).

## Animal perfusion and tissue preparation for imaging

At the end of ischaemia/reperfusion animals were overdosed with pentobarbital sodium and trans-cardially perfused with saline (200 ml) followed by 4% paraformaldehyde (PFA, 200 ml) for fixation and then 20 ml of 5% gelatin (Sigma-Aldrich, G2625) solution containing FITC-albumin conjugate (Sigma-Aldrich, A9771) to enable visualization of the perfused coronary microvasculature. The hearts were then fixed overnight in 4% PFA, and 150 μm transverse sections made for immunofluorescence staining. Pericytes were labelled with anti-NG2 antibodies (Merck Millipore) and the capillary basement membrane and pericytes were labelled (*Mishra et al., 2014*) with isolectin $B_4$-Alexa Fluor 647 (Molecular Probes, I32450).

The capillary diameters we measure with this protocol are not affected by vessels being compressed by myocyte contraction. The initial perfusion with calcium-free saline stops the heart in diastole. Evidence that the heart is indeed arrested in diastole is provided by the large volume visible within the left ventricle (*Figure 2A,C*), which matches that seen during diastole in magnetic resonance images of rat heart, which also show that the volume during systole is far smaller (see Figure 2B of *Crowley et al., 1997*). Indeed, it is clear that the observation of occluded capillaries after ischaemia does not reflect compression by arrest in systole, because essentially no capillary block was observed in control hearts that were not made ischaemic (only 3% of capillaries were blocked, compared to 40% after ischaemia: see *Figure 2D*). Another demonstration that the observation of occluded capillaries does not reflect compression by arrest in systole is provided by the fact that capillary occlusion (*Figure 2D*) and reduction of FITC-albumin labelling (*Figure 2B*) were only seen on the side of the heart where the blood supply was transiently interrupted, and not on the normally perfused side of the heart, and were specifically associated with pericytes (*Figure 3H*). Furthermore, our mean capillary diameter in control hearts (5.38 ± 0.28 microns, see *Figure 3J*) is similar to that estimated for capillaries in diastole in rat hearts (measured in relaxed hearts as 5.3 microns with a suggested correction to 5 microns: see *Henquell et al., 1976*) and is greater than the diameter estimated for systole (~4 microns: *Henquell et al., 1976*).

## Imaging of vessels after ischaemia experiments

Image z-stacks of overall left ventricular myocardial blood volume, the perfusion of capillaries in the area at risk, and individual vessel blockages, were acquired using laser scanning confocal microscopy (Zeiss LSM 700), and analysed using ImageJ software (NIH, Bethesda, MD, USA). To quantify blood volume across the left ventricular myocardium,~30 low power z-stacks were taken of an entire transverse section of each heart (using a 5X air objective), maximum intensity projected, and stitched

together using the MosaicJ plugin (*Thévenaz and Unser, 2007*) of ImageJ (n = 6 hearts for ischaemia, n = 8 for ischaemia +adenosine, n = 5 for control). Usually some FITC-albumin labelled blood remained within the lumen of the ventricles (although this blood often became detached during tissue processing). For ease of interpretation of the images in *Figure 2*, this labelling (and also the area outside the imaged heart) was digitally recoloured black using Photoshop. All quantification of the FITC labelling was carried out on the unedited images.

For quantification of the percentage of capillaries that were perfused, three randomly selected regions of the anterior left ventricle wall were imaged in both ischaemia and sham animals, as this region of myocardium consistently included the ischaemic area (which showed visible pallor and oedema of the myocardium). The person quantifying the images was blinded to the condition that the heart was exposed to. 75 stacks (160 µm square, and 20 µm deep) in 12 different animals were taken (ischaemia n = 4, ischaemia +adenosine n = 5, control n = 3). Blockages of flow in the ischaemic area at risk were identified by abrupt terminations in FITC-IB$_4$ signal. For ischaemia 42 blockages were imaged; for ischaemia +adenosine 14 blockages were imaged. Clots (or red blood cell rouleaux) were very rarely observed.

## Image analysis

For low power blood volume analysis, 12 regions of interest (ROI) were drawn clockwise around the left ventricle (when looked at from above) from the mid-point of the septum (as in *Figure 2A*), and the mean intensity of FITC-albumin signal was recorded for each ROI, and normalised to the highest intensity measured in any ROI. These data were averaged over hearts and renormalized so that the mean value in positions 1–3 of *Figure 2C* was 1. This signal is assumed to be proportional to the volume of blood perfusing the myocardium. To compare perfusion in the ischaemic zone, we averaged the plotted values over ROIs 7–10 for each heart in the different conditions, and compared the mean values averaged over all the hearts studied in each condition.

To quantify the percentage of perfused capillaries, we counted the number of filled and unfilled vessels crossing a line drawn through the centre of each image perpendicular to the main capillary axis.

For blockages in flow, we measured the distance along the capillary from the termination of the FITC-albumin signal to the mid-point of the nearest visible pericyte soma on the same capillary. The cumulative probability distribution of the blockage-to-nearest-soma distance (over all blocked capillaries) was compared with the distribution predicted for random placement of a blockage on each blocked capillary. If pericytes were a uniform 60 µm distance apart (*Figure 1F*) along unbranched capillaries, then the latter cumulative probability distribution for the blockage-to-nearest-soma distance would increase linearly with distance, to reach a value of 1 at a distance of 30 µm (half the distance between two pericytes). In practice, pericytes vary in their separation, vessels branch and, because it was necessary to image at high magnification to reliably locate blockage sites, sometimes the blocked vessel passes out of the image before one of the pericytes adjacent to the block is reached (making it uncertain whether all positions on the imaged capillary are in fact closest to the imaged pericyte, rather than one outside of the image). We calculated the expected cumulative probability distribution of the distance from imaged pericytes to a randomly positioned vessel blockage as follows. For each blocked capillary the probability, p(x).dx, of a location on the capillary (and hence a potential blockage location) existing between distances x and x + dx from the pericyte soma (with x $\geq$ 0) was calculated as a function of distance from the soma (see *Figure 3—figure supplement 1*). For example, for a capillary of length L in the image, with a single pericyte soma located a distance A from one end, the probability is

p(x).dx=2.dx/L for x < A,
p(x).dx=dx/L for A < x < L-A,
and p(x).dx=0 for x > L A,

with the total probability (integrated along the capillary length) summing to 1. For more complicated geometries (e.g. two pericytes on the blocked capillary, or branching capillaries) the same approach was used to calculate p(x), taking into account all possible positions on the capillaries. The resulting probability distribution was then averaged over all 42 images used for *Figure 3H*. To generate a predicted cumulative probability distribution with the same resolution as that obtained for the experimentally observed blockages, this smooth distribution was then sampled at 42 equi-

probability points as shown in *Figure 3H*.The resulting distribution shows a deviation from a linear increase to a value of 1, at 30 µm distance, because of the factors mentioned above.

## Statistics

Statistical analysis employed OriginPro. Data normality was assessed with Shapiro-Wilk tests. Comparisons of normally distributed data were made using 2-tailed Student's t-tests. Equality of variance was assessed with an F test, and heteroscedastic t-tests were used if needed. Data that were not normally distributed were analysed with Mann-Whitney tests. P values were corrected for multiple comparisons using a procedure equivalent to the Holm-Bonferroni method (for N comparisons, the most significant p value is multiplied by N, the 2nd most significant by N-1, the 3rd most significant by N-2, etc.; corrected p values are significant if they are less than 0.05). Cumulative probability distributions were compared using the Kolmogorov-Smirnov test. An estimate of the sample size needed for a typical experiment is as follows: For a control response of 100%, a response standard deviation of 25%, a response after a manipulation (ischaemia or adenosine) of 50% (50% inhibition), a power of 80% and p<0.05, 6 samples are needed (http://www.biomath.info/power/ttest.htm) in each of the control and manipulation groups. The exact numbers depend on the effect size for the manipulation and the standard error of the data.

## Acknowledgements

This work was supported by the Rosetrees Trust, European Research Council, Fondation Leducq, Wellcome Trust, Medical Research Council, and a Marie Curie Fellowship of the EU (654691 to SM). We thank IS Cohen, MC Ford, A Gourine, CN Hall, C Howarth, R Jolivet, A Mishra, EA Newman, R Nortley and DM Yellon for comments on the manuscript.

## Additional information

### Funding

| Funder | Grant reference number | Author |
| --- | --- | --- |
| H2020 European Research Council | BrainPower | David Attwell |
| Rosetrees Trust | A1188 | Fergus M O'Farrell David Attwell |
| H2020 Marie Sk?odowska-Curie Actions | 654691 | Svetlana Mastitskaya |
| Fondation Leducq | 08CVD02 | David Attwell |
| Wellcome | 075232 | David Attwell |

The funders had no role in study design, data collection and interpretation, or the decision to submit the work for publication.

### Author contributions

Fergus M O'Farrell, Matthew Hammond-Haley, Conceptualization, Data curation, Formal analysis, Investigation, Visualization, Methodology, Writing—original draft, Writing—review and editing; Svetlana Mastitskaya, Conceptualization, Resources, Data curation, Formal analysis, Investigation, Methodology, Writing—original draft, Writing—review and editing; Felipe Freitas, Conceptualization, Formal analysis, Investigation, Visualization, Methodology, Writing—review and editing; Wen Rui Wah, Investigation, Visualization; David Attwell, Conceptualization, Resources, Data curation, Formal analysis, Supervision, Funding acquisition, Investigation, Writing—original draft, Project administration, Writing—review and editing

### Author ORCIDs

Fergus M O'Farrell http://orcid.org/0000-0002-7378-9175
David Attwell http://orcid.org/0000-0003-3618-0843

## Ethics

Animal experimentation: Experiments were performed in accordance with European Commission Directive 2010/63/EU (European Convention for the Protection of Vertebrate Animals used for Experimental and Other Scientific Purposes) and the UK government Animals (Scientific Procedures) Act (1986), with project approval from the UCL Animal Welfare and Ethical Review Body.

## Decision letter and Author response

Decision letter https://doi.org/10.7554/eLife.29280.009
Author response https://doi.org/10.7554/eLife.29280.010

## Additional files

### Supplementary files

• Transparent reporting form
DOI: https://doi.org/10.7554/eLife.29280.007

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
