## [Decision Letter]

Thank you for submitting your article "Capillary pericytes mediate coronary no-reflow after myocardial ischaemia" for consideration by *eLife*. Your article has been reviewed by three peer reviewers, one of whom, Reinhard Fässler, has agreed to reveal his identity. The evaluation has been overseen by Fiona Watt as the Senior Editor and Reviewing Editor.

The reviewers have discussed the reviews with one another and the Reviewing Editor has drafted this decision to help you prepare a revised submission.

O'Farrell et al. have examined a possible role for microvascular pericytes in post-cardiac ischemia lack of capillary reperfusion, or "no-reflow". The question is important and of high medical significance. The report is well written and, overall, well illustrated. The first part of the work is a description of vascular cell types in the rat myocardium after lectin (IB4) binding and surface marker visualisation, by IHC or in transgenic reporter mice.

Next, rat heart ischemia was created by coronary artery ligation, and no-reflow documented in the microvasculature after systemic injection of FITC-albumin. Precise statistical comparison between sites of blood flow interruption and pericyte position, correlated with reduction in vessel diameter, led the authors to conclude that lack of capillary reperfusion was a consequence of pericyte constriction. In agreement with this, administration of adenosine, a pericyte relaxant, reduced the number of blocked capillaries and, consequently, improved reperfusion. This extends to the cardiac microvasculature observations made previously by the same group on brain blood vessels and may indicate cardiac pericytes as therapeutic target cells in cardiac ischemia.

Essential revisions

1) In the reported experiments, capillary diameter reduction resulting from pericyte constriction is of the order of 37%, which is not necessarily sufficient to interrupt the blood flow (or, at least, the plasma flow) unless, as suggested by the authors, blood cells cannot crawl through the constriction and eventually completely clog the conduit. This latter point should be investigated by immunohistochemical detection of glycophorin A and CD45, for instance, on sections such as those shown in Figure 3. In addition, there is no clear indication if the heart was perfused with a solution that allows stopping the heart in diastole. This condition is essential to avoid the possibility that some vascular areas are occluded by contraction.

2) It is possible that the differences in cell death of pericytes in the present paper and the earlier Nature paper would disappear if identical occlusion and reperfusion times had been used. The authors need to examine this possibility experimentally.

3) Not all pericytes express α-SMA, although the distribution of contractile and non-contractile pericytes among venules, arterioles and capillaries and in different organs remains to be precisely described. Immunohistochemical detection of α-SMA expression by pericytes adjacent to capillary blockage spots would add strength to the paper, even more so if more distant pericytes are found to be α-SMA negative. In addition, the definition of pericytes requires additional staining. How do the authors exclude the possibility that some NG2-cells are vascular muscle cells?

4) Adenosine is a widely used vasodilator, acting via e.g. A2 receptors to dilate smooth muscle cells. If A2 receptors are expressed on pericytes, they would dilate, too. The primary effect of adenosine is exerted on smooth muscle cells both preclinically and in preconditioning scenarios such as reactive hyperemia. The claim that adenosine works predominantly on pericytes needs more convincing evidence. In addition, the mechanism of pericyte contraction is hypothetical. The authors should test whether neuroadrenergic blockade reverses the phenomenon.

Other comments

1) The paper contains nice images showing pericytes encircling capillary vessels. Less expected is the observation that some vessels are devoid of pericyte coverage (Figure 1). Could these be very small, or tangentially sectioned venules (hence associated with NG2-negative pericytes)? This point deserves comments in the discussion.

2) Important control experiments, in which pericyte functions are abrogated (e.g. via Angiopoietin-2) are missing.

3) The authors do not provide adequate context for their observations in terms of the pathophysiology of no-reflow, which requires a plaque, an occluding thrombus, a mechanical removal of the clot by balloon inflation and an activated, edematous microvascular endothelium. In addition, the roles of invading neutrophils and platelets are not discussed. The study is conducted in mice and the limitation of the model should be considered, as the human vasculature is different from rodents in several important respects.

---

## [Author Response]

Essential revisions1) In the reported experiments, capillary diameter reduction resulting from pericyte constriction is of the order of 37%, which is not necessarily sufficient to interrupt the blood flow (or, at least, the plasma flow) unless, as suggested by the authors, blood cells cannot crawl through the constriction and eventually completely clog the conduit. This latter point should be investigated by immunohistochemical detection of glycophorin A and CD45, for instance, on sections such as those shown in Figure 3.

As the referees note, ischaemia-evoked pericyte constriction reduces the capillary diameter by 37% to 3.4 microns at pericyte locations (Figure 3). However, this value is for *all* pericyte somata, not just those that evoked capillary block (and furthermore the block often did not occur exactly at the pericyte soma), so this value is not the diameter associated precisely with the location of blockages. We have now explained more clearly in the legend to Figure 3 that it is the mean diameter at all pericyte somata that is plotted, and mentioned that if only pericytes associated with blockages are selected then the diameter is less (3.19 ± 0.24 μm, n=30). Nevertheless, even this reduction in diameter is apparently not sufficient to completely occlude the vessel, which raises the referee’s question of whether cells get stuck at the regions of reduced diameter.

As is now explained in subsection “Pericytes constrict capillaries after ischaemia”, since leukocytes are both larger than erythrocytes (8.5 microns compared with 7.2 microns in diameter: Komatsu et al., 1990, Microvasc Res 40, 1) and less deformable than erythrocytes (Schmid-Schonbein et al., 1981, Biophys J 35, 243; Downey et al., 1985, J Appl Physiol 69, 1767; Doerschuk et al., 1993, J Appl Physiol 74, 3040), we expected that leukocytes rather than red blood cells would get stuck at capillary regions constricted by pericytes. As suggested by the referee, we attempted to demonstrate leukocyte or red blood cell trapping at sites of capillary constriction by pericytes.

Firstly, we examined labelling with antibody to neutrophil elastase or ICAM-1 (since granulocytes may block coronary capillaries and cause no-reflow: Engler et al., (1983, 1986), but see Habazettl et al., (1999, references in our paper). Surprisingly, although we could detect leukocytes outside vessels (Figure 3; usually post-capillary venules but also some capillaries), we observed none in the lumen at 46 sites of capillary block. On the other hand, when we labelled red blood cells with antibody to glycophorin A, as suggested, we did observe that some blockages had red blood cells stuck in them, but this was not a universal feature of blockages (only 18% of 44 blockages were occluded by red blood cells) and the presence of a red blood cell at a constriction did not guarantee a blockage of blood flow.

We have now described these data in subsection “Pericytes constrict capillaries after ischaemia”2, as follows:

“Surprisingly, labelling with antibody to neutrophil elastase or ICAM-1 revealed no leukocytes lodged at 46 blockage sites examined (although, as a positive control, they were seen outside vessels, Figure 3, usually post-capillary venules). Similarly, labelling for the erythrocyte protein glycophorin A revealed red blood cells (Figure 3) associated with only a small percentage of blockage sites (18% of 44 blockages), and even where red blood cells were trapped at capillary constrictions it did not always lead to a block of blood flow (as shown by FITC-albumin passing the red blood cells)”.

We have also added, as Figure 3a picture of red blood cells occluding a capillary at a pericyte location and an image of a blockage lacking neutrophil elastase labelling but showing a neutrophil outside a capillary. Furthermore, in the Discussion section, we have now speculated that:

“The fact that we observed few cells in the capillary lumen at blockages may result from their displacement during the perfusion with FITC-albumin in gelatin, or may alternatively reflect capillary block occurring when constriction brings together the layers of glycocalyx (Secomb et al., 1998) on opposite sides of the capillary.”.

In addition, there is no clear indication if the heart was perfused with a solution that allows stopping the heart in diastole. This condition is essential to avoid the possibility that some vascular areas are occluded by contraction.

The referees are referring to the fact that, on each cardiac cycle, blood flow through the cardiac capillaries is prevented in systole because of the contraction of the myocardium. There are 5 lines of evidence implying that this does not affect our assessment of capillary block by pericyte constriction, as follows.

i) First, the heart is stopped in diastole by the perfusion of calcium-free saline, before perfusion of the green (FITC-albumin) label used to define perfused and non-perfused vessels.

ii) Evidence that the heart is indeed arrested in diastole is provided by the large volume visible within the left ventricle (Figure 2), which matches that seen during diastole in magnetic resonance images of rat heart, which also show that the volume during systole is far smaller (see Figure 2 of Crowley et al., 1997, Exp Physiol 82, 887).

iii) It is clear that the observation of occluded capillaries after ischaemia does not reflect compression by arrest in systole, because essentially no capillary block was observed in control hearts that were not made ischaemic (only 3% of capillaries were blocked, compared to 40% after ischaemia: see Figure 2).

iv) Another demonstration that the observation of occluded capillaries does not reflect compression by arrest in systole is provided by the fact that capillary occlusion (Figure 2) and reduction of FITC-albumin labelling (Figure 2) were only seen on the side of the heart where the blood supply was transiently interrupted, and not on the normally perfused side of the heart, and were specifically associated with pericytes (Figure 3).

v) Finally, our mean capillary diameter in control hearts (5.38+0.28 microns, see Figure 3) is similar to that estimated for capillaries in diastole in rat hearts (measured in relaxed hearts as 5.3 microns with a suggested correction to 5 microns: see Henquell et al., 1976, Microvasc Res, 12, 259) and is greater than the diameter estimated for systole (~4 microns: Henquell et al., 1976, Microvasc Res, 12, 259).

We have now explained all this in the Materials and methods section.

2) It is possible that the differences in cell death of pericytes in the present paper and the earlier Nature paper would disappear if identical occlusion and reperfusion times had been used. The authors need to examine this possibility experimentally.

In fact, we did not investigate pericyte death in the current paper, but merely mentioned that the significant reversal of part of the 37% post-ischaemic decrease in capillary diameter at pericyte locations (Figure 3) seen on adding adenosine after the ischaemia indicates that not all the pericytes can be dead. The duration of ischaemia we use in these experiments (45 mins) is in fact included in the chemical ischaemia protocol that we used for brain pericytes (Figure 4C of Hall et al., 2014, Nature 508, 55), where it led to the death of approximately 50% of pericytes (on the y axis of that figure, 3.3 dead pericytes/100 microns is equivalent to 100% death, which took just over an hour). The fact that, in the current work, adenosine only allowed partial (and not complete) recovery of the decrease in flow would be consistent with some pericytes having died in rigor at the end of the ischaemia (in which case the percentage dead could be similar to that induced by chemical ischaemia in brain slices).

We have now discussed all this in the Discussion section of the paper and stated that adenosine might produce a greater restoration of flow if pericyte death were prevented.

3) Not all pericytes express α-SMA, although the distribution of contractile and non-contractile pericytes among venules, arterioles and capillaries and in different organs remains to be precisely described. Immunohistochemical detection of α-SMA expression by pericytes adjacent to capillary blockage spots would add strength to the paper, even more so if more distant pericytes are found to be α-SMA negative.

The heterogeneity of a-SMA expression that the referee mentions has led to suggestions that, for the CNS, it might be γ-actin rather than a-SMA that mediates contraction. We have now carried out further experiments to assess which actin isoform might mediate the contraction of pericyte processes that underlie capillary constriction after cardiac ischaemia. First, we examined the fraction of pericytes that express different smooth muscle actin isoforms. The results are shown in Figure 1 and described in subsection “Pericytes associate with capillaries and sympathetic axons, and express actin”,as follows:

“Pericytes are conventionally assumed to constrict capillaries using α-smooth muscle actin (α-SMA: Joyce et al., 1985; Skalli et al., 1989), but variability in the labelling observed for α-SMA and data showing expression of other actin isoforms in pericytes has led to a suggestion, for CNS pericytes, that γ-actin might instead be the relevant actin isoform (DeNofrio et al., 1989; Grant et al., 2017). We therefore examined antibody labelling for α-SMA, β-actin and γ-actin (Figure 1). α-SMA labelling occurred in 36.4 ± 6.7% of 57 pericytes (Figure 1), and was rarer than labelling for β-actin (62.3 ± 2.8% of 30 pericytes) or γ-actin (76.0 ± 3.4% of 47 pericytes). α-SMA-expressing pericytes were most commonly observed in capillaries closer to the arteriole end of the capillary bed, while β- and γ-actin were seen in pericytes across the capillary bed.”

We find that after ischaemia the capillary diameter at pericyte somata is reduced more near blockages (reduced to 3.19 ± 0.24 μm as mentioned above) than at non-block locations (3.64 ± 0.23 μm), but even the latter diameter is reduced from that in control conditions (5.38 ± 0.28 μm, significantly different, p=1.56x10^-5^). This implies that it is not the case that only a small fraction of pericytes – for example those that express α-SMA – constrict capillaries in ischaemia. Instead they may all constrict, with the blockages occurring at the sites of greatest constriction. Examining actin isoform labelling of pericytes located at blockages, we obtained the following results, which are now described in subsection “Pericytes constrict capillaries after ischaemia***”***and shown in Figure 3:

“Pericytes near blockage sites were tested for labelling of the different actin isoforms mentioned above. All 4 such pericytes tested for α-SMA labelling exhibited labelling (Figure 3), as did 5 out of 6 tested for β-actin, and all 6 tested for γ-actin. Since only a small fraction of pericytes may need to constrict a capillary to abolish its blood flow, further work is needed to determine which is the main actin isoform responsible for the ischaemia-evoked contraction of pericyte processes and reduction of blood flow.”

In addition, the definition of pericytes requires additional staining. How do the authors exclude the possibility that some NG2-cells are vascular muscle cells?

The main markers used to define pericytes in most studies are NG2 and PDGFRβ. As requested, we have now carried out further labelling, and added images to Figure 1 showing that the cells we are studying express both NG2 and PDGFRβ. Furthermore, we have now quantified overlap of the expression of these markers in left ventricular cardiac pericytes in adult NG2-DsRed mice. In the left ventricle, 99.2 ± 0.2% of NG2-expressing pericytes also expressed PDGFRβ (2852 of 2874 cells analysed in 36 z-stacks from 4 mice). Furthermore, 93.6 ± 1.0% of PDGFRβ-expressing pericytes also expressed NG2 (2852 of 3047 cells analysed in 36 z-stacks from 4 mice). This value is significantly different to 100% (p=2.3x10^-7^), suggesting that there may be a small population (6.4%) of PDGFRβ-expressing pericytes not expressing NG2.

These results have now been described in subsection “Pericytes associate with capillaries and sympathetic axons, and express actin” and the legend to Figure 1, and shown in Figure 1.

The referee questions the distinction between pericytes and vascular smooth muscle cells (SMCs). A similar question has recently arisen for pericytes in the CNS, and been assessed in depth (Attwell et al., 2016, JCBFM 36, 451) in the light of the historical definition of pericytes. In brief, the distinction between pericytes and SMCs is an anatomical one. SMCs consist of immediately adjacent band-like cells running around arterioles, while pericytes have spatially-separated somata (located either with a bump-on-a-log appearance on the straight parts of capillaries, or at capillary branch points) and are found all along the capillary bed. It is important to note that pericytes may differ in the morphology of their processes (and possibly also in the proteins they express) at different positions along the capillary bed but, nevertheless, the historical definition of pericytes encompassess all of these forms of the cells (Attwell et al., 2016, JCBFM 36, 451). The anatomical definition of pericytes has now been stated on page 4, para 1, of the manuscript.

4) Adenosine is a widely used vasodilator, acting via e.g. A2 receptors to dilate smooth muscle cells. If A2 receptors are expressed on pericytes, they would dilate, too. The primary effect of adenosine is exerted on smooth muscle cells both preclinically and in preconditioning scenarios such as reactive hyperemia. The claim that adenosine works predominantly on pericytes needs more convincing evidence.

We are certainly not claiming that adenosine acts only on pericytes, and it is indeed likely to also relax arteriolar smooth muscle cells (an effect we now mention in subsection “Adenosine reduces pericyte constriction and no-reflow“). However, we are claiming that adenosine can reduce no-reflow after cardiac ischaemia by relaxing pericytes. Indeed, we already provide direct evidence in the paper for adenosine relaxing pericytes. In Figure 3 we show that i) after ischaemia, the capillary diameter at pericyte locations is reduced by 37%; and ii) after ischaemia plus adenosine on reperfusion, the capillary diameter is reduced by only 23%.

Thus, adenosine has dilated the pericytes.

Evidence that this is not a general effect on the capillaries, unrelated to the pericytes, is provided in Figure 3, where we show that the effects of ischaemia and of adenosine are specific to the location of the pericyte soma, and are not seen 10 microns away.

We have now stressed in the Abstract that the capillary dilation produced by adenosine occurs specifically at pericytes. We have also discussed this in subsection “Adenosine reduces pericyte constriction and no-reflow” –, where we state:

“This increase of flow could reflect adenosine acting both on arteriolar smooth muscle and on pericytes. However, adenosine also reduced by one quarter the percentage of capillaries that were blocked after reperfusion, from ~40% to ~30% (p=0.007, Figure 2). […] Since the capillary blockages induced by ischaemia in the absence of adenosine are disproportionately associated with pericytes, and since the adenosine was only applied around the period of reperfusion, these data suggest that, at least in part, adenosine reduces no-reflow by reversing the constriction of pericytes that ischaemia induces. To test this hypothesis, we compared the diameter of capillaries at pericyte somata with the diameter 10 μm upstream of the soma and found that adenosine significantly (p=0.0045) reduced the constriction evoked by ischaemia at pericyte somata (Figure 3), implying a specific effect on pericytes (rather than a general capillary dilation produced by upstream arteriole dilation). The absolute capillary diameter at pericyte somata after ischaemia was increased by 21% using adenosine (p=0.025, Figure 3). Thus, adenosine decreases no-reflow by relaxing pericytes.”

In addition, the mechanism of pericyte contraction is hypothetical. The authors should test whether neuroadrenergic blockade reverses the phenomenon.

We expect that, as for brain pericytes (Hall et al., 2014), the loss of energy supply in ischaemia will lead to a rise of [Ca^2+^]_i_ which evokes contraction, as stated at the start of the Discussion. However, noradrenaline is released in ischaemia (as we already stated in the Discussion) so, as requested, we assessed the effect of blocking vasoconstrictive α_1_ receptors with terazosin. This had no effect on the percentage of capillaries remaining blocked after ischaemia. These data are described insubsection “Pericytes constrict capillaries after ischaemia” –, where we state:

“We found no effect of injecting terazosin (0.5 mg/kg i.v., 5 mins before occluding the LAD artery, which produced a 20-33 mm Hg decrease of blood pressure), with 48.8 ± 3.7% blocked in the presence of terazosin (in 13 images covering 383 capillaries in 2 hearts; not significantly different from the 44.7 ± 4.9% seen for 14 images covering 366 capillaries in 2 hearts in interleaved experiments without terazosin, p=0.52).”

Other comments1) The paper contains nice images showing pericytes encircling capillary vessels. Less expected is the observation that some vessels are devoid of pericyte coverage (Figure 1). Could these be very small, or tangentially sectioned venules (hence associated with NG2-negative pericytes)? This point deserves comments in the discussion.

In general, the cardiac capilaries are fairly uniformly invested with pericytes, as shown in Figure 1. It is true that Figure 1 (previously Figure 1) shows a few vessels with stretches of 40 microns that appear to have no pericytes (in fact there are some pericyte processes present on two of the four vessels that we believe the referee means, but they are labelled too dimly to be visible without over-saturating the representation of the brighter-labelled pericytes in the picture). The vessels shown were completely within the confocal planes imaged, and so do not reflect tangential sectioning of vessels. They are also unlikely to be venules because of their small diameter (<4 microns for the least labelled vessels in Figure 1). However, with a mean distance between pericyte somata of almost 60 microns (Figure 1), random variation will result in occasional capillary lengths of 40 microns that lack pericyte somata and processes (indeed we already discussed the effect of this variation in the Image Analysis section of the Materials and methods section). As requested, we have now added a comment on this in subsection “Pericytes associate with capillaries and sympathetic axons, and express actin”, where we report the mean distance between pericytes.

2) Important control experiments, in which pericyte functions are abrogated (e.g. via Angiopoietin-2) are missing.

We agree that it would be of interest to investigate whether deletion of signalling via the angiopoietin/TIE and PDGF/PDGFRbeta pathways affect the contractile properties of cardiac pericytes and their response to ischaemia. These pathways have been shown to regulate pericyte adhesion to vessels and capillary permeability, both in the CNS (Gurnik et al., 2016, Acta Neuropathol 131, 753; Part et al., 2016, Nat. Commun. 8, 15296; Daneman et al., 2010, Nature 468, 562; Bell et al., 2010 Neuron 68, 409; Armulik et al., 2010, Nature 468, 557) and in the heart (Ziegler et al., 2013, PMID: 23863629), but it is unknown whether loss or over-expression of this signalling affects expression of contractile proteins by pericytes. We estimate that it would take about 2 years to conduct such a study, given the need to (i) transfer our ischaemia techniques from rat to mouse, (ii) generate colonies of mice with molecules deleted specifically from cardiac pericytes or endothelial cells (global deletion is not sufficient as it modifies peripheral resistance: Ziegler et al., 2013 PMID: 23863629), and (iii) carry out the experiments. We believe these experiments are therefore outside the scope of this study, but we have now added a sentence in the Discussion section suggesting such an approach for the future and citing the Ziegler et al., (2013) paper.

3) The authors do not provide adequate context for their observations in terms of the pathophysiology of no-reflow, which requires a plaque, an occluding thrombus, a mechanical removal of the clot by balloon inflation and an activated, edematous microvascular endothelium. In addition, the roles of invading neutrophils and platelets are not discussed. The study is conducted in mice and the limitation of the model should be considered, as the human vasculature is different from rodents in several important respects.

We have now added a paragraph dealing with this in the Discussion section.